# A junction coverage compatibility score to quantify the reliability of transcript abundance estimates and annotation catalogs

Charlotte Soneson[1,2], Michael I Love[3,4], Rob Patro[5], Shobbir Hussain[6], Dheeraj Malhotra[7], Mark D Robinson[1,2]

**Most methods for statistical analysis of RNA-seq data take a matrix of abundance estimates for some type of genomic features as their input, and consequently the quality of any obtained results is directly dependent on the quality of these abundances. Here, we present the junction coverage compatibility score, which provides a way to evaluate the reliability of transcript-level abundance estimates and the accuracy of transcript annotation catalogs. It works by comparing the observed number of reads spanning each annotated splice junction in a genomic region to the predicted number of junction-spanning reads, inferred from the estimated transcript abundances and the genomic coordinates of the corresponding annotated transcripts. We show that although most genes show good agreement between the observed and predicted junction coverages, there is a small set of genes that do not. Genes with poor agreement are found regardless of the method used to estimate transcript abundances, and the corresponding transcript abundances should be treated with care in any downstream analyses.**

## Introduction

High-throughput sequencing of the transcriptome (RNA-seq) is used for a broad range of applications in biology and medicine. Most of these involve comparing expression levels of genetic features (e.g., genes, transcripts, or exons) between samples, and the quality of the results from any such study will therefore be directly dependent on the correctness of the expression estimates for the particular features of interest. The ability to obtain accurate estimates, in turn, depends on the quality and quantity of the available data and the completeness and correctness of the used reference annotation. In general, reliable abundance estimation is

easier to achieve for genes than for individual transcripts or isoforms because of high sequence similarity among groups of isoforms and the nonuniform read coverage resulting from library preparation and sequencing biases (Kanitz et al, 2015; Soneson et al, 2015). However, gene-level abundance estimation is not without challenges, particularly for groups of genes that share a large fraction of their sequence, which leads to high numbers of multi-mapping reads (Paşaniuc et al, 2011; Robert & Watson, 2015; McDermaid et al, 2018 Preprint). Various solutions have been proposed, including grouping together similar genes (Robert & Watson, 2015), probabilistic assignment of reads to genes (Paşaniuc et al, 2011), and scoring the genes based on their sequence similarity and number of multi-mapping reads shared with other genes (McDermaid et al, 2018 Preprint).

Despite their higher reliability, gene-level abundances are insufficient for analyses aimed at detecting differences in transcript-level expression or relative isoform usage. Even for studies where the main aim is to detect differential expression at the gene level, incorporating transcript abundances can in some cases improve the inference (Wang et al, 2010; Trapnell et al, 2013; Soneson et al, 2015). As methods for transcript abundance estimation are improving, both in accuracy and speed, it has become increasingly common to estimate abundances of individual isoforms rather than of the gene as a whole, and today a plethora of transcript abundance estimation methods based on various underlying algorithms are available (e.g., Trapnell et al, 2010; Li & Dewey, 2011; Glaus et al, 2012; Roberts & Pachter, 2013; Patro et al, 2014; Lee et al, 2015; Pertea et al, 2015; Bray et al, 2016; Liu & Dickerson, 2017; Patro et al, 2017). Most evaluations of the ability of these methods to accurately estimate transcript abundances have been performed using simulated data, where reads are generated from a known transcriptome (Kanitz et al, 2015; Soneson et al, 2015), or using artificial spike-in sequences (Leshkowitz et al, 2016). Evaluations have also been performed based on the agreement of abundance estimates between replicates (Teng et al, 2016) or agreement with abundances

[1]Institute of Molecular Life Sciences, University of Zurich, Zurich, Switzerland    [2]SIB Swiss Institute of Bioinformatics, University of Zurich, Zurich, Switzerland    [3]Department of Biostatistics, University of North Carolina-Chapel Hill, Chapel Hill, NC, USA    [4]Department of Genetics, University of North Carolina-Chapel Hill, Chapel Hill, NC, USA    [5]Department of Computer Science, Stony Brook University, NY, USA    [6]Department of Biology and Biochemistry, University of Bath, Bath, UK    [7]F. Hoffmann-La Roche Ltd, Pharma Research and Early Development, Neuroscience, Ophthalmology and Rare Diseases, Roche Innovation Center Basel, Basel, Switzerland

Correspondence: charlotte.soneson@fmi.ch; mark.robinson@imls.uzh.ch
Charlotte Soneson's present address is Friedrich Miescher Institute for Biomedical Research and SIB Swiss Institute of Bioinformatics, Basel, Switzerland

or abundance ratios derived from other types of data such as exon arrays (Dapas et al, 2016), RT-PCR (Zhang et al, 2015), or 3′ end sequencing (Kanitz et al, 2015). Less is known about the reliability of transcript abundance estimates in real data sets, based on potentially inaccurate or incomplete annotation catalogs, and how to spot unreliably quantified transcripts in a sample-wise manner based on the RNA-seq data alone. A motivating example is illustrated in Fig 1A, showing abundance estimates for the *ZADH2* gene in Epstein-Barr virus (EBV)-transformed lymphocytes, as displayed in the Genotype-Tissue Expression (GTEx) Portal (https://www.gtexportal.org/home/gene/ZADH2, accessed July 19, 2018). This gene has four annotated isoforms, each consisting of two exons and each featuring a unique splice junction (with a shared acceptor site). The top row illustrates the estimated expression of collapsed exons and junctions (with legends to the right), indicating a high expression of the most 5′ exon and the corresponding junction. The alternative exons and junctions

have no or very few supporting reads. However, the isoform abundance estimates (lower panel) suggest a different picture, where two of the isoforms whose unique exons and junctions are supported by few reads are assigned the highest expression levels.

In this article, we present the junction coverage compatibility (JCC) score (Fig 1B), which allows detection of genes with such conflicting indications of isoform abundance. The score can be calculated for any genomic region (e.g., a gene locus), by comparing the observed coverage profile, obtained by aligning the RNA-seq reads to the genome, with the predicted coverage profile derived from estimates of transcript abundances and biases influencing the observed read coverage of a sequenced transcript. In particular, we focus on the number of reads spanning annotated splice junctions in the genomic region of interest. The key assumption behind the JCC score is that with (i) a complete and accurate catalog of reference transcripts, (ii) an accurate estimate of the abundance of

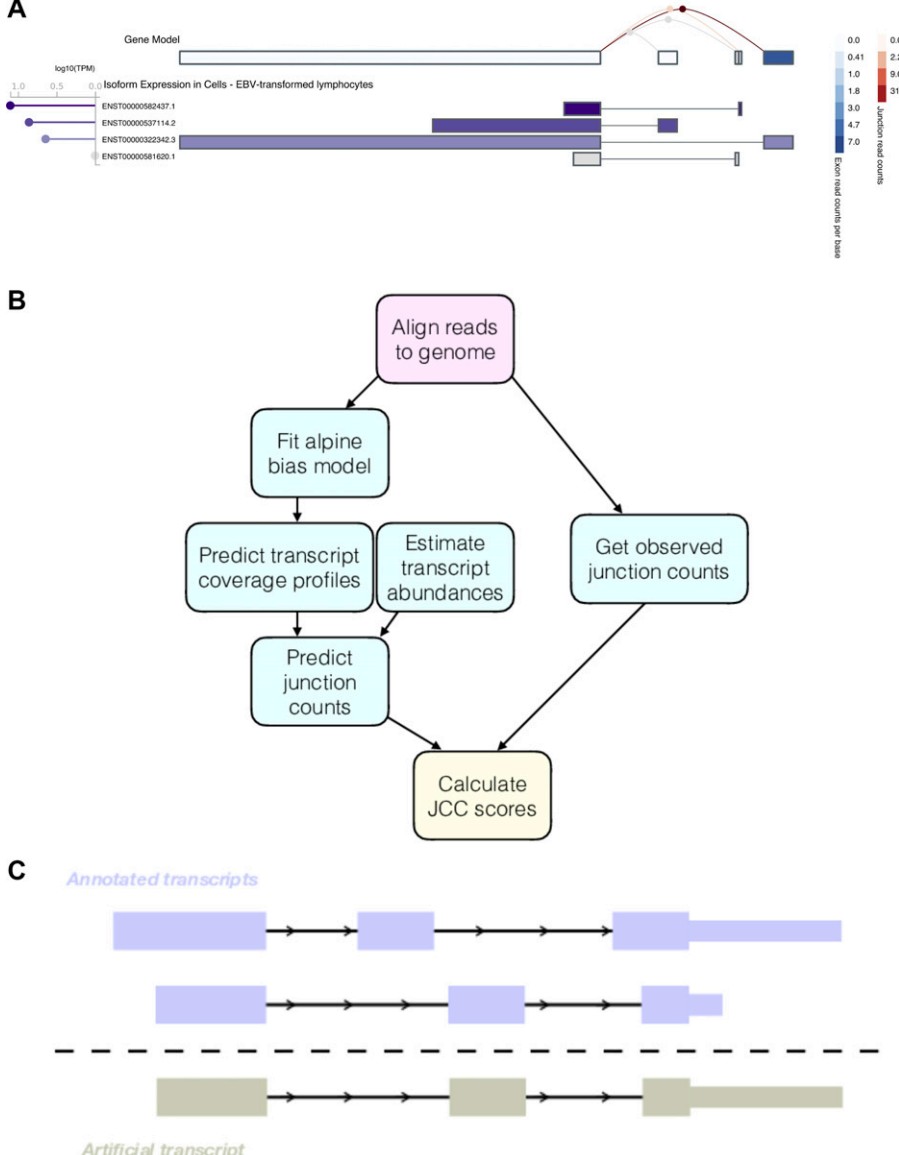

**Figure 1. Motivation and outline of the JCC score. (A)** Example of a gene with inconsistent signals resulting from abundance estimation based on exons, junctions, or entire isoforms. The figure was generated in the GTEx Portal (https://www.gtexportal.org/home/gene/ZADH2, accessed July 19, 2018). **(B)** Outline of the approach used to calculate the JCC scores. First, reads are aligned to the genome using STAR, and the number of reads observed to span each annotated splice junction is extracted. The aligned reads are also used to fit a fragment bias model using the alpine Bioconductor package, which is then used to predict coverage profiles for all annotated transcripts. The coverage profiles are combined with transcript abundance estimates to obtain the predicted numbers of junction-spanning reads, which are compared with the observed numbers to calculate the JCC score for each gene. **(C)** Schematic illustrating the generation of artificial transcripts in the simulated data. In total, artificial transcripts are generated for 4,514 genes, which have multiple annotated 3′UTR of different length (at least 1-kb length difference) starting in the same genomic position. For each such gene, two transcripts are selected; one that is annotated with the short 3′UTR and one that is annotated with the long one. The artificial transcript is created by combining the internal structure (all exonic regions except the annotated 3′UTR) of one of the two isoforms with the 3′UTR of the other. In the simulation, all reads from the modified genes are generated from the artificial transcripts.

each individual transcript, and (iii) knowledge about the biases affecting the probability of a given fragment of a given transcript to be sequenced, the coverage profile prediction obtained by combining these three sources of information for any genomic locus should be close to the observed one. Thus, large deviations between the observed and predicted coverage profiles indicate that the transcript estimates in the region are unreliable, and such regions should be flagged and interpreted with caution in downstream analyses. There can be many reasons behind a region obtaining a high (bad) JCC score, ranging from poor performance of the estimation method, for example, due to sequence similarity with other parts of the transcriptome or low read coverage of regions critical for distinguishing transcripts, to an incorrect or incomplete annotation catalog, making a correct distribution of the reads between the annotated transcripts in the region impossible.

Using eight transcript abundance estimation methods and two deeply sequenced human RNA-seq data sets (denoted as *Cortex* and *HAP1*, see the Materials and Methods section), we show that for most human genes, the junction coverages predicted from the transcript abundances are highly concordant with the observed junction coverages, suggesting overall accurate annotation and transcript abundance estimates. However, a small fraction of the annotated genes show a substantial difference between the predicted and observed junction coverages. For some of these genes, the reason for the incompatibility appears to be an incompletely annotated transcript catalog, and no distribution of the reads among the annotated isoforms would simultaneously give a satisfactory JCC and a good agreement with the annotated UTRs. The uneven read coverage of isoforms also leads to estimation problems, especially for genes with short, poorly covered exons. Using a simulated data set, we show that misannotation of 3'UTRs can lead to unreliable transcript estimates, which is interesting in the light of recent reports showing that most isoform differences between tissues are due to alternative start and end sites and involve untranslated exons (Pal et al, 2011; Shabalina et al, 2014; Reyes & Huber, 2018).

# Results

### Predicted transcript coverage patterns agree well between samples

The prediction of the transcript coverage profiles by alpine is a crucial step in the calculation of the JCC score. It is carried out separately for the *HAP1* and *Cortex* samples, to account for any sample-specific biases. Of the 200,310 annotated transcripts in the Ensembl GRCh38.90 gtf file, the prediction of the coverage pattern by alpine failed for 29,342 (14.6%) in the *HAP1* sample and 13,906 (6.9%) in the *Cortex* sample, almost exclusively because of transcripts being shorter than the respective fragment lengths. The prediction returned NULL for 23,028 (11.5%) transcripts in the *HAP1* sample and 11,941 (6.0%) in the *Cortex* sample that did not have any overlapping reads. For these transcripts, we impose a uniform coverage, rather than excluding them from subsequent calculations.

Overall, we observe a high correlation between the predicted coverage profiles in the two libraries (Fig S1), indicating that they share many of the biases, despite coming from different cell types and being prepared and sequenced almost two years apart on different sequencing machines. The coverage prediction is the single most time-consuming step of the JCC score calculation, and the high correlation even between such different libraries suggests that in a specific study, the prediction may not need to be done separately for each individual sample, which can reduce the run time considerably. Run time can also be reduced by limiting the coverage prediction and subsequent analysis to transcripts from a subset of the genes that are of particular interest in a given situation.

### Most predicted junction coverages are consistent with the observed coverages

Using the approach described in the Materials and Methods section, we obtain the number of uniquely mapping reads observed to span each annotated junction and the number predicted to span each junction given each set of transcript abundance estimates. Comparing the predicted junction coverages ($C_j$) with the observed ones ($R_j$) across all annotated junctions shows a generally high correlation for all abundance estimation methods (Fig 2A, left column), suggesting that in most genomic loci, the annotated transcript structure is compatible with the observed read alignments and that the approach we use to predict junction coverages based on transcript abundances is valid. Scaling the predicted junction coverages within each gene, corresponding to setting $\beta = 1$ in the subsequent JCC calculation (see the Materials and Methods section) and thereby focusing more on the relative junction coverages within a gene rather than the overall abundance of the gene, increases the correlation for all methods (Fig 2A, right column). The largest discrepancies between observed and predicted junction coverages are seen for SalmonCDS, indicating that on a global scale, only considering annotated coding sequences discards relevant information about transcript abundances. We also note that there is a set of junctions with a low fraction of uniquely mapping reads (Fig 2A, marked in red) for which the predicted number of spanning reads is considerably higher than the observed number of uniquely mapping junction reads. Because these discrepancies do not represent a failure of the annotation system or transcript abundance estimation method, but rather an inability to place reads in a unique genomic position, we downweight the influence of these junctions on the gene-wise JCC score via the $g(\omega)$ function, as described in the Materials and Methods section. Permuting the transcript counts within each gene leads to substantially lower correlations (Fig S2), suggesting that the high correlation is not driven mainly by the expression level of the genes, but by a correct distribution of reads among isoforms.

### Most genes show high compatibility between observed and predicted junction coverages

After investigating the concordance between observed and predicted coverages for individual junctions, we next calculate the JCC score for each annotated gene. With the exception of SalmonCDS (which is using a reference annotation in which many transcripts and genes are missing because they do not have an explicitly

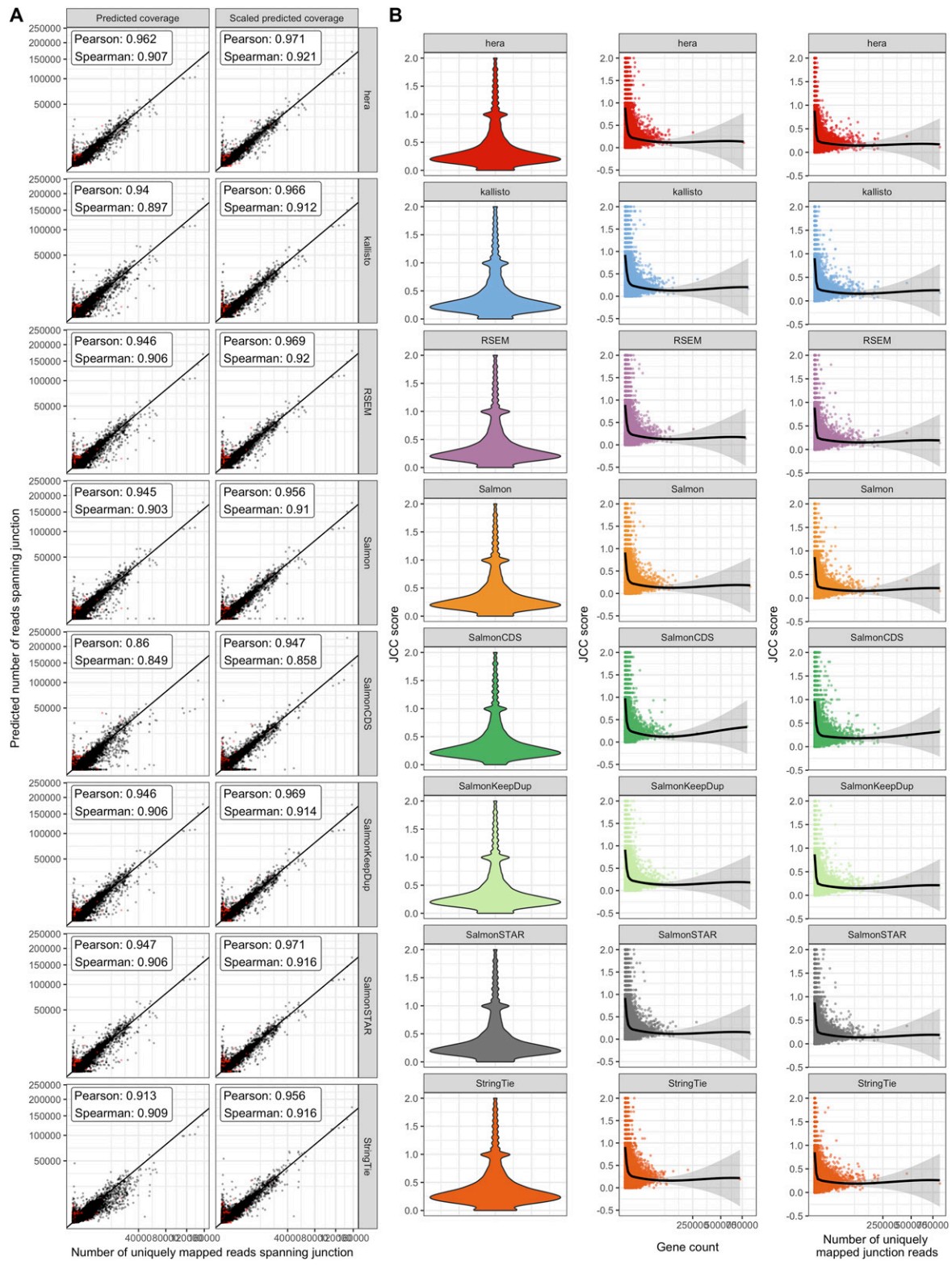

**Figure 2. Comparison of observed and predicted coverage patterns. (A)** Correlation between observed and predicted number of reads spanning each junction for the *HAP1* sample. The left column ("Predicted coverage") shows the actual number of reads predicted by alpine and the respective transcript abundance estimation method, whereas the predicted values in the right column ("Scaled predicted coverage") are scaled to sum to the same number as the observed number of uniquely mapping junction reads within each gene. Scaling improves the correlation between observed and predicted junction coverages for all included methods. Axes are square root transformed for better visualization. Red points indicate junctions where less than 75% of the spanning reads are uniquely mapping. **(B)** Overall distribution of the gene-wise JCC scores for each method in the *HAP1* sample, as well as the association between the JCC score and the total number of reads for the gene and the number of uniquely mapped junction reads in the gene.

annotated coding sequence), we are able to calculate a valid JCC score for around 16,500 genes in the *HAP1* library and just more than 20,000 genes in the *Cortex* library (Fig S3). Among the genes for which the score cannot be calculated, most are not expressed (predicted total abundance of all isoforms is equal to 0), whereas a smaller fraction either are expressed but lack junctions, or contain junctions but have no or too few uniquely mapping junction-spanning reads to calculate the score.

Investigating the overall distribution of valid JCC scores shows that for most genes, the score is low (below 0.5), confirming the previous observation that for most of the genes, the junction coverage pattern induced by the estimated transcript abundances agrees well with the observed junction coverages (Fig 2B, left column). Similar distributions are seen for all included abundance estimation methods; in particular, genes with high JCC scores are observed with all abundance estimation approaches. Most of the very high scores are obtained for genes with low abundance and few uniquely mapped reads spanning any of the junctions (Fig 2B). The high score for these genes may be driven largely by shot noise and may improve with even higher sequencing depth. Moreover, lowly expressed genes are typically excluded in practical analyses of RNA-seq data such as differential expression analyses. Thus, to illustrate the behaviour of the JCC score, in the following analyses, we focus on genes with at least 25 reads mapping uniquely across any of its junctions.

### JCC scores are overall similar between methods

Because the JCC score is obtained by combining a set of estimated transcript coverage profiles with transcript abundance estimates, using different transcript abundance methods for the latter leads to different sets of scores. We calculate JCC scores using transcript abundance estimates from eight different methods, and subsequently calculate correlation coefficients between the scores obtained by each method pair, using only genes with at least 25 uniquely mapping junction-spanning reads (Figs S4, S5, and S6). As expected, the correlation is overall very high, and the most deviating scores are obtained with SalmonCDS, which uses a different set of reference sequences than the other methods, and StringTie. On average, both SalmonCDS and StringTie give higher scores than the remaining methods (Fig S6B).

### Examples of genes with high JCC scores

To exemplify the types of deviating patterns resulting in high JCC scores, we consider some of the genes that are assigned high scores (JCC ≥ 0.6) with all the transcript abundance methods (except SalmonCDS, because it is based on a different set of reference transcripts and does not represent a typical or recommended way of performing transcript abundance estimation). The rationale for focusing on these genes is that we expect genes that are consistently assigned a high score, regardless of the way the transcript abundances were estimated, to be more likely to harbor misannotated transcripts or suboptimal read coverage patterns, making abundance estimation difficult. For genes where some abundance estimates provide compatible junction coverage patterns, high scores for other methods are more likely due to

problems in the abundance estimation step. Furthermore, we limit the investigation to genes with at least 25 uniquely mapped junction-spanning reads, at least 75% of the junction-spanning reads mapping uniquely and an intron/exon read count ratio below 0.1. These strict filtering criteria are satisfied by 161 genes in the *Cortex* library and 58 genes in the *HAP1* library. Eighteen of the genes pass the filters in both libraries. One of these genes is *ZADH2* (Fig 3). *ZADH2* has four annotated transcripts, each consisting of two exons and one exon–exon junction, and no junction is shared between transcripts. Most transcript abundance estimation methods distribute the estimated abundance between two or three of these isoforms. However, only one of the four annotated junctions has any observed spanning reads, which suggests that only the corresponding transcript (ENST00000322342) is indeed present. This leads to a large discrepancy between the observed and predicted junction coverages (for all abundance estimation methods), and hence a large JCC score. For this gene, a possible explanation for the discrepancy is that the coverage of the 5′ end of the transcripts is weak, but for a reason not captured by the alpine bias model, implying that the 3′ end, which is longer and shows a higher coverage, will dominate the abundance estimation. Uneven coverage in this region can, therefore, bias the abundance estimation towards one or the other transcript. As illustrated in Fig 1A, a similar behaviour can be seen also in the GTEx data (accessed via the GTEx Portal).

Investigation of the 18 genes that received high scores with all quantification methods in both samples suggests that they can be broadly divided into three groups. The first group consists of genes similar to *ZADH2*, where a low or uneven coverage of the 3′ and/or 5′ end of transcripts leads to a read assignment that is incompatible with the observed junction coverage pattern (for other examples, Figs S7, S8, S9, and S10). The second group of genes obtaining high JCC scores across methods and data sets are those where the annotation catalog appears to be incomplete, or where the annotated 3′UTRs are seemingly too short (examples in Figs S11 and S12). Finally, the third group consists of a small set of genes where the reason for the high score is unclear from visual inspection because of complicated transcript configurations and uneven coverage patterns (Figs S13 and S14). Taken together, these observations support the hypothesis that high JCC scores that persist across several different abundance estimation approaches and multiple data sets are more likely to be caused by transcriptome misannotation rather than imperfections in the abundance estimation procedure itself. Regardless of the cause, however, the resulting abundances are unreliable and should be interpreted with caution in downstream analyses. We also note that because the JCC score depends not only on the annotation catalog but also on the estimated abundances, even incorrectly annotated genes will only be assigned a high JCC score for samples where unannotated transcripts are indeed expressed.

### JCC scores are not strongly associated with inferential variability

Several isoform abundance estimation methods allow assessment of the variability of the resulting expression levels via some form of (re)sampling (Li & Dewey 2011; Glaus et al, 2012; Turro et al, 2014; Bray et al, 2016; Mandric et al, 2017; Patro et al, 2017). To compare the

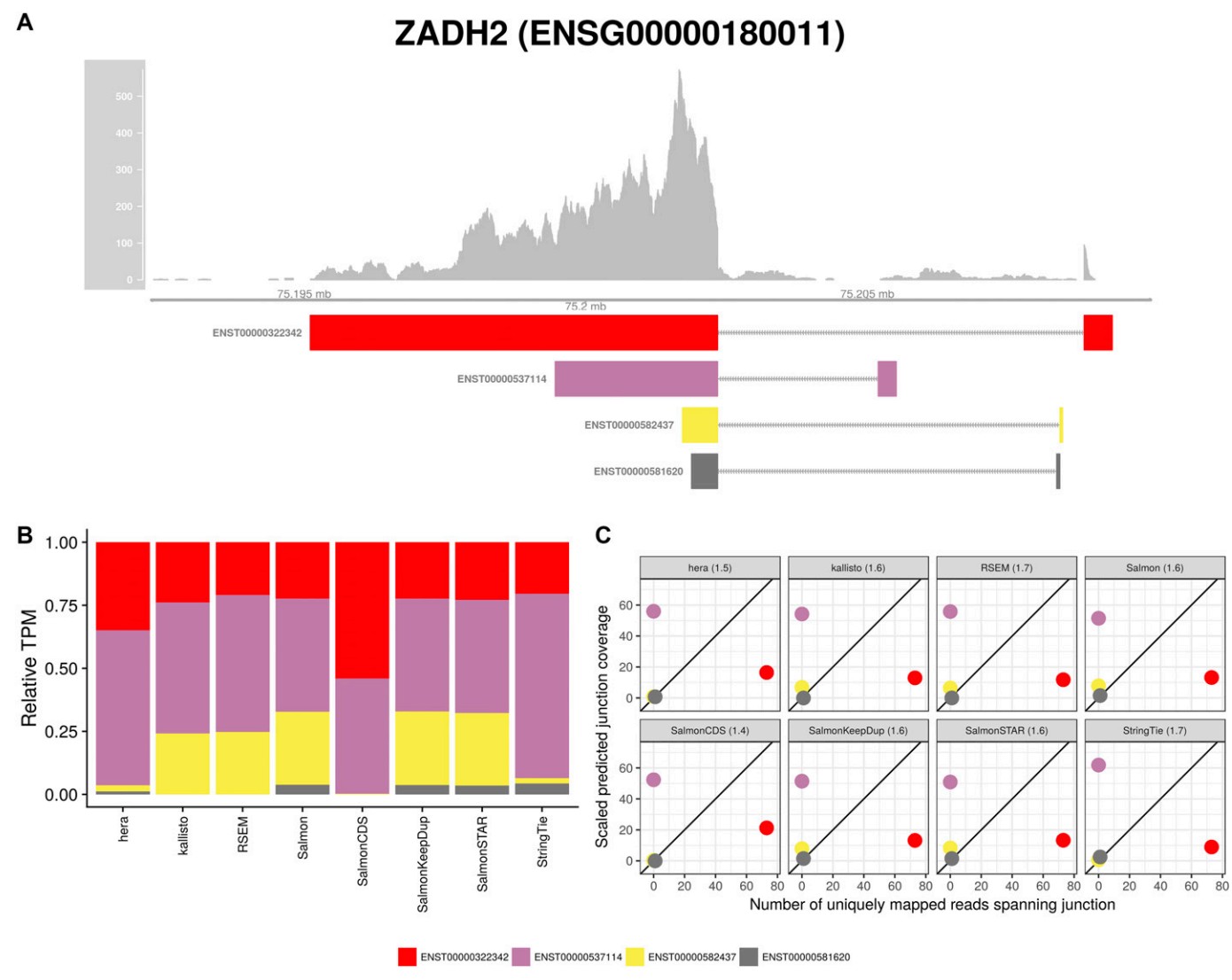

**Figure 3.  Estimated abundances and JCC scores for the *ZADH* gene. (A)** Observed coverage profile and annotated gene model for the *ZADH2* gene in the *HAP1* library. Different annotated transcripts are shown in different colors. **(B)** Relative TPM estimates for the annotated transcripts from each of the eight transcript abundance estimation methods. **(C)** Observed number of uniquely mapping junction-spanning reads (*x*) and scaled predicted junction coverages (*y*) based on transcript abundance estimates from each of the eight methods. Each circle corresponds to an annotated junction and is colored according to the set of transcripts that it is annotated to. The JCC scores for this gene based on the abundances from the respective abundance estimation approaches are indicated in the panel headers.

uncertainties picked up by the JCC score with those represented in these inferential variances, we perform 100 bootstrap runs using Salmon and estimate the coefficient of variation of the boot-strapped counts both at the transcript level and after aggregating the transcript counts at the gene level. For the evaluation, we consider only genes with at least 25 uniquely mapping junction-spanning reads, and each individual transcript is assigned the JCC score of the corresponding gene. Overall, the association between the inferential coefficient of variation and the JCC score is weak in both libraries, at both the transcript and gene level (Fig S15). Thus, the two scores measure different types of uncertainties; although the bootstrap variability may capture assignment uncertainty caused by shared sequence features among transcripts, it will not in general pick up inconsistencies due to misannotation, which are targeted by the JCC score.

## The choice of reference annotation affects the JCC score distribution

All analyses reported previously were performed using the Ensembl GRCh38.90 annotation. To investigate the impact of the choice of reference annotation on the JCC scores, we estimate bias models and predict transcript coverage profiles also for all transcripts in the CHESS 2.0 catalog (Pertea et al, 2018). We estimate corre-sponding transcript abundances with Salmon and kallisto and count junction-spanning reads for each annotated junction with STAR. The CHESS catalog was obtained by assembling reads from almost 10,000 GTEx samples and contains a larger number of transcripts (annotated to a smaller number of genes) than the Ensembl catalog (Table S1). The CHESS genes are all annotated with a unique CHESS identifier, but a mapping to Entrez IDs is provided

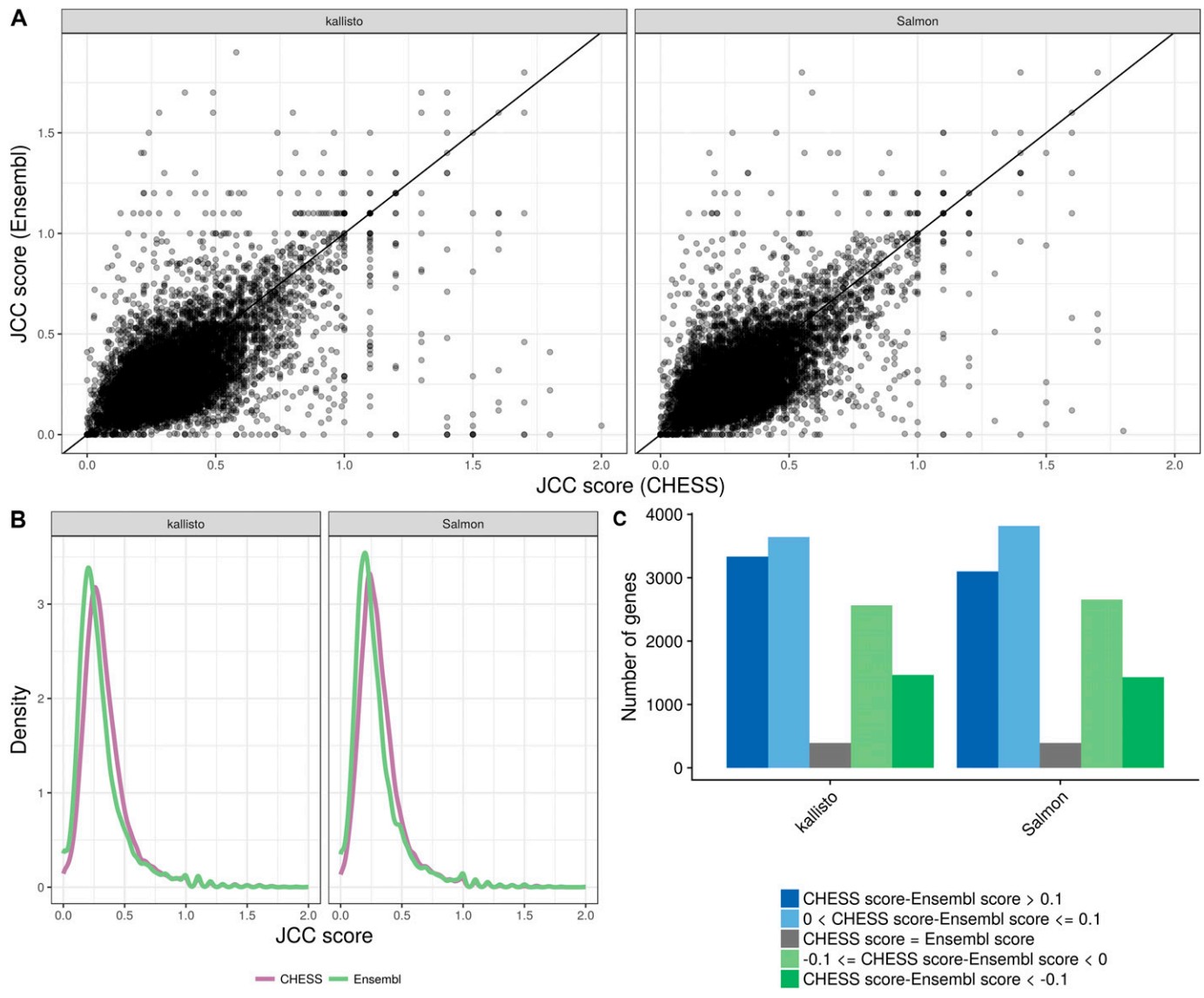

**Figure 4. Comparison between scores obtained with the Ensembl GRCh38.90 annotation and the CHESS 2.0 annotation, for the *HAP1* sample.**
**(A)** Correlation between scores obtained with the CHESS annotation (*x*) and the Ensembl annotation (*y*), for all the shared genes (genes with an assigned Ensembl ID in the CHESS catalog), with at least 25 uniquely mapping junction-spanning reads and at least 75% of the junction-spanning reads mapping uniquely with both annotations. **(B)** Distribution of JCC scores for all genes with at least 25 uniquely mapping junction-spanning reads and at least 75% of the junction-spanning reads mapping uniquely, in the respective annotation catalogs. **(C)** The number of genes shared between the two annotation catalogs for which the CHESS annotation results in a higher, lower, or equal score compared with the Ensembl annotation. Blue bars represent genes for which scores based on the CHESS annotation are higher (worse) than those based on the Ensembl annotation and green bars represent the opposite situation.

wherever possible. For comparison with our other results, we convert the Entrez IDs to Ensembl IDs using the org.Hs.eg.db Bioconductor package v3.6.0 (in this way, unique Ensembl IDs are obtained for 22,262/42,881 = 51.9% of the genes). Considering only genes that are shared between the two annotation catalogs, it is clear that there is a substantial difference between the scores assigned to an individual gene using the two annotations (Fig 4A), although the overall distribution of scores is largely similar (Fig 4B). Neither annotation catalog is consistently leading to lower scores than the other (Fig 4C), but there are genes with substantially lower scores with each of the two annotations compared with the other.

In addition, we investigate the effect of quantifying transcript abundances using a data set–specific catalog of transcripts, obtained by running StringTie (without the -e argument) on each of the two Illumina libraries. The resulting gtf file contains many new transcripts, and many annotated transcripts from the Ensembl catalog are removed (Table S1). We apply a subset of the abundance estimation methods to the respective StringTie annotations and compare JCC scores across all genes present in both the StringTie and Ensembl catalogs. Also in this case, no annotation consistently lead to lower scores than the other, but there is a larger fraction of genes that show lower scores with the sample-specific StringTie-assembled annotation (Fig S16).

**Misannotated 3′UTRs strongly affect the abundance estimates**

To investigate the effect of misannotated or missing 3′UTRs on the transcript abundance estimates, and consequently the JCC score, in more detail, we used synthetic data. For each of 4,514 annotated genes, we generated an artificial transcript consisting of the coding sequence of one isoform and the 3′UTR of another isoform from the same gene. The two contributing isoforms were selected in such a way that one was annotated with a short 3′UTR, and the other with a long 3′UTR (with a length difference of at least 1 kb) starting in the same genomic location. As expected, for genes where the isoform with the long 3′UTR was selected to contribute the 3′UTR to the artificial transcript, a large fraction of the final artificial transcript consists of the 3′UTR, whereas the fraction is much smaller if the 3′UTR was chosen from the isoform with the short 3′UTR (Fig S17).

For the modified genes, reads are simulated only from the artificial transcript. We also simulate reads from a random selection of unmodified transcripts. As expected, the JCC scores for the genes with modified transcripts are generally higher than those for the genes without any modified transcripts, where the reads are simulated from the correct annotation catalog (Fig S18A). The distribution of scores for the latter set of genes can be seen as a "baseline distribution" of scores that we can expect for reasons unrelated to annotation and sequencing artifacts (e.g., sequence similarity causing problems for abundance estimation methods). Furthermore, the JCC score is generally higher for genes where a larger fraction of the artificial transcript is made up of the 3′UTR (Fig S18B). Focusing only on the genes with modified transcripts, we calculate the similarity between the artificial transcript and all annotated transcripts from the same gene. The similarity is defined by the Jaccard index of the nucleotide positions covered by the two compared transcripts. We stratify the genes based on whether the most similar transcript to the artificial transcript is the one that contributed the internal structure, the one that contributed the 3′UTR, or another one of the annotated transcripts. For most abundance estimation methods, the annotated transcript that is most similar to the artificial transcript (from which the reads were generated) is also assigned the highest expression estimate (Fig 5). The exceptions are SalmonCDS and StringTie, which both generally assign the highest abundance to the transcript that is most similar to the artificial transcript in terms of the internal structure, rather than based on overall similarity. This is consistent with the observation described previously that SalmonCDS and StringTie tended to provide different scores than the other methods.

To further investigate incompatible junction coverage patterns induced by misannotated 3′UTRs in the experimental data, we generate an extended transcript catalog by expanding each explicitly annotated 3′UTR to include the longest annotated 3′UTR starting in the same position. The resulting transcript is added to the original set of Ensembl transcripts, with a suffix "longUTR" added to the original identifier. A somewhat similar approach was taken in a previous study (Zhang et al, 2017), which noted that variations in the 5′ and 3′ ends of transcripts from *Arabidopsis thaliana* can affect abundance estimation and alternative splicing identification, and that padding of the 5′ and 3′ ends of transcripts before transcript abundance estimation resulted in improved

correlation with splicing ratios from HR RT-PCR. Rerunning the JCC score estimation with the expanded Ensembl catalog led to a lower score for a set of genes, and a higher score for others (Fig S19). The latter may potentially be explained by the increased redundancy in the expanded catalog and illustrates that a more extensive transcript catalog does not automatically lead to improved abundance estimates. Focusing on the genes for which the JCC score is consistently improved with the expanded catalog, across abundance estimates from different methods, we could indeed identify genes where the distribution of reads was largely driven by a long 3′UTR rather than adherence to internal JCC, and where extending the 3′UTR of transcripts with a compatible junction chain improved the read assignment and thereby led to a lower JCC score (Figs S20, S21, S22, S23, S24, S25, S26, S27, S28, and S29).

## Discussion

We have described the JCC score and shown how it can be used to identify genes or genomic regions where junction coverage patterns predicted from estimated transcript abundances are incompatible with those observed after alignment of the RNA-seq reads directly to the genome. By using the RNA-seq data to obtain two estimates of the number of reads mapping across each splice junction, we can create an internal validation system, thereby circumventing the need for an external data set or additional replicates for evaluation of transcript abundance estimation accuracy. A high score, indicating poor compatibility between the junction coverages estimated from the transcript abundance estimates and the observed junction coverages, can be caused, for example, by inaccurate transcript abundance estimates (e.g., for transcripts that share large parts of their sequence with other transcripts) or by an incomplete or incorrect transcriptome annotation. Regardless of the underlying cause, such genes should be flagged in downstream analyses and the estimated transcript abundances interpreted with caution. We note that the results were overall similar for all the eight transcript abundance estimation approaches used in the study, representing alignment-free methods and methods relying on either genome or transcriptome alignments.

The chosen reference annotation can have a large effect on the resulting JCC scores, as seen here by comparing the scores obtained using the Ensembl annotation to those based on the CHESS 2.0 annotation. In addition, using StringTie to assemble missing transcripts led to improved scores for a large number of genes and a worse score for a smaller number of genes. As recommended (https://github.com/alexdobin/STAR/blob/2.5.3a/doc/STARmanual.pdf), we used the primary genome assembly from Ensembl for aligning the reads to the genome. However, the transcriptome FASTA files from Ensembl, which were used as the basis for abundance estimation by Salmon, SalmonKeepDup, kallisto, RSEM, and SalmonSTAR, contain transcripts from alternative contigs that are not included in the primary genome assembly. Many of these transcripts are identical or very similar to transcripts annotated to the primary chromosomes. Although this represents the typical use of these

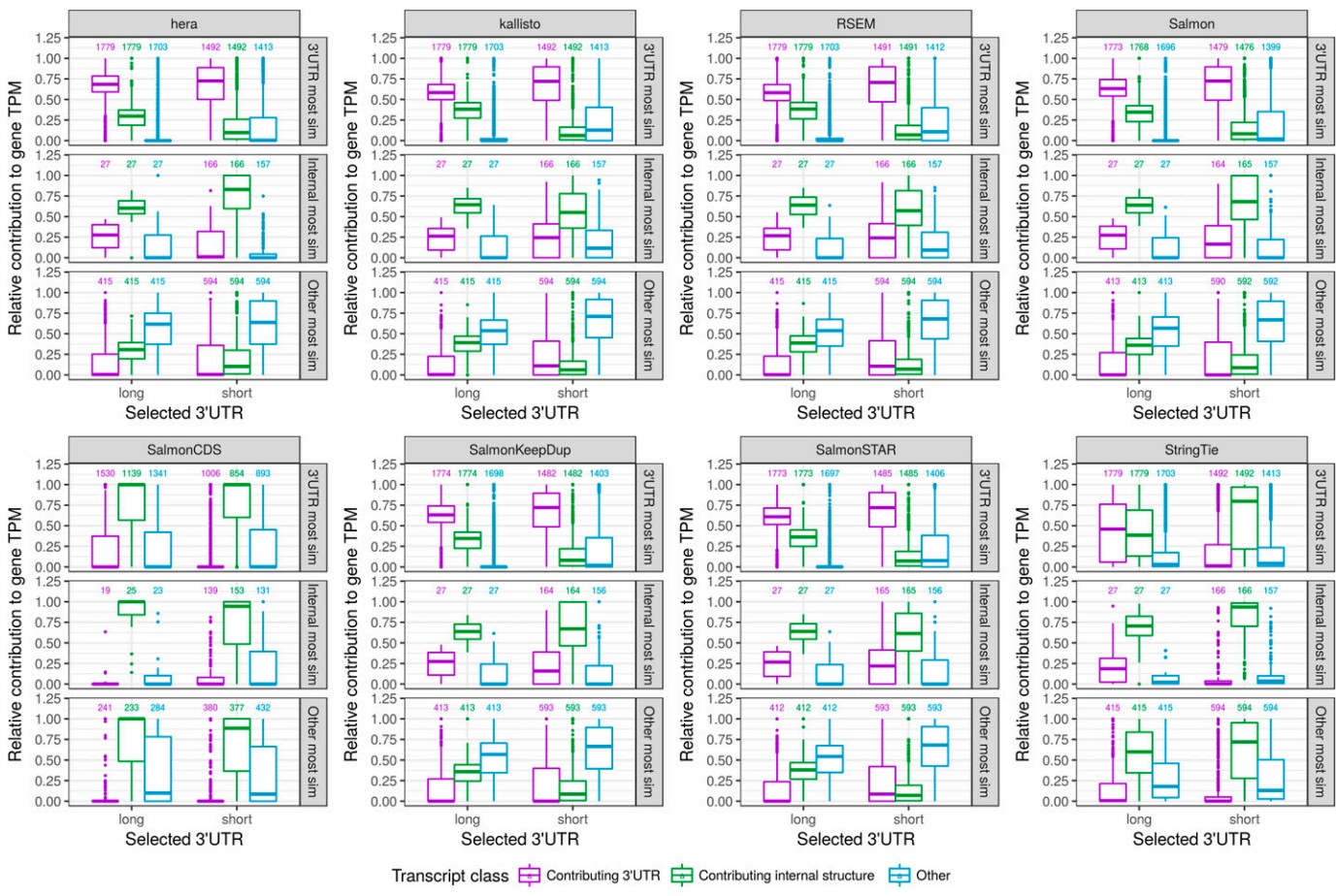

**Figure 5. Relative transcript abundances for modified genes in the simulated data set, with each of the eight transcript abundance estimation methods.**
Genes are stratified (vertically) based on whether the transcript that is most similar (by Jaccard index of covered nucleotides) to the artificial transcript is the one contributing the 3'UTR, the one contributing the internal structure, or another isoform of the gene (see Fig 1C). For each gene, we calculate the relative abundance of the transcript contributing the 3'UTR, the one contributing the internal structure, and all other isoforms of the gene combined (indicated with color). Finally, the genes are stratified (horizontally) based on whether the artificial transcript contains the long or short variant of the 3'UTR. Generally, most methods assign the highest abundance to the transcript that is most similar to the artificial transcript from which the reads were generated, with the exception of SalmonCDS and StringTie, which assign higher abundances to the transcripts that are most similar to the artificial transcript in terms of the internal structure. The numbers above the boxplots indicate the number of genes in each category.

alignment files for alignment and transcript abundance estimation, it may lead to problems for the correct assignment of the reads to transcripts, and as a consequence, for the calculation of the JCC scores. Keeping only one representative of duplicate transcript sequences (the default behaviour of Salmon) can lead to both better abundance estimates and improved agreement between predicted and observed junction coverages, under the assumption that the correct transcript location is retained. Of course, determining the true location of origin of a given transcript can be highly nontrivial, but would be an interesting direction for future research.

One limitation of the presented family of JCC scores is that they cannot be calculated for genes that do not have annotated junctions or that do not have reads spanning junctions. A solution to this could be to compare the predicted and observed coverage profiles of the entire genomic locus rather than just the junctions. However, multi-mapping reads will still pose a problem for the comparison, and positions with a large fraction of multi-mapping overlapping reads should be downweighted in the score. In general,

the approach we propose is not limited to junction coverages and could be extended to, for example, disjoint exon bins. The requirement is that we can observe the coverage pattern of the features of interest from the genome alignment and predict it from the alpine bias models and the estimated transcript abundances. In addition, although we use the weighting function $g(\omega)$ to downweight the influence of junctions with a large fraction of multi-mapping reads, it can be used more generally to assign weights to junctions based on any characteristics affecting our confidence in the observed read coverages.

Our evaluations are based on the assumption that we are interested in obtaining and using transcript abundance estimates. Other quantification approaches, for example, those focusing on disjoint exon bins (Anders et al, 2012) or transcript equivalence classes (Ntranos et al, 2016) have been suggested, and the resulting counts may in themselves be less sensitive to uncertainties in the reference transcript catalog. However, a post-processing step is required to interpret the results in terms of known transcripts, and

during this step, misannotated transcripts can still lead to erroneous conclusions.

Using simulated data, we observed that compared with the other abundance estimation methods, StringTie appeared to focus more on matching the internal structure than the 3′UTR when assigning abundances to transcripts. This implies that in situations where the 3′UTR annotation is unclear, StringTie can help assigning the reads to the transcript that is most similar with respect to the more unambiguous part of the transcript structure. However, it could potentially also make it more difficult to identify differences in transcript composition between tissues because these have been shown to be predominantly different in the transcription start and end sites (Reyes & Huber, 2018).

Our results show that for the vast majority of the human genes, the junction coverage patterns implied by the estimated transcript abundances in our data sets agree well with the observed ones, indicating that the reference annotation and transcript abundance estimates for these genes are likely to be reliable. However, for each transcript abundance estimation method, a small number of genes obtained a high JCC score, suggesting unreliably quantified isoforms. These genes should be treated with care in any downstream analyses or be investigated further for an improved transcriptome annotation.

# Materials and Methods

## Experimental data and reference annotations

We use two deeply sequenced human polyA+ RNA-seq libraries for our investigations. The first (*Cortex*) contains 117,292,547 paired-end 126-nt Illumina reads from a human cerebral cortex sample and the second (*HAP1*) contains 55,234,720 paired-end 151-nt Illumina reads from the HAP1 cell line. Both samples were prepared with the Illumina TruSeq RNA-stranded protocol and sequenced at the Functional Genomics Center in Zurich, Switzerland; *Cortex* with a HiSeq 2500 in October 2015 and *HAP1* with a HiSeq 4000 in September 2017. Most of our analyses are performed using the GRCh38.90 reference annotation from Ensembl (Zerbino et al, 2018). For comparison, we also use the recent CHESS 2.0 reference catalog (Pertea et al, 2018), which was generated by assembling RNA-seq reads from almost 10,000 GTEx samples (GTEx Consortium, 2013; Carithers et al, 2015) using StringTie (Pertea et al, 2015). Based on the original Ensembl gtf file, we generate two additional gtf files, containing flattened exonic regions and intronic regions (regions within a gene locus that are not covered by any exon) and use featureCounts (Liao et al, 2014) (from subread v1.6.0; [Liao et al, 2013]) to count the number of reads overlapping these exonic and intronic regions for each gene.

## Simulated data

In addition to the experimental RNA-seq data sets, we generate synthetic data with the aim to better understand the effect of misannotated 3′UTR sequences. From the GRCh38.90 Ensembl annotation, we find 4,514 genes with multiple annotated 3′UTRs starting in the same position, and with length difference exceeding 1 kb. For each of these genes we randomly extract one transcript annotated with the short 3′UTR and one transcript annotated with the long one. We then generate an artificial transcript, consisting of the 5′UTR and coding sequence of one of these two transcripts and the 3′UTR of the other transcript (Fig 1C). For 41 of the 4,514 genes (0.9%), the artificial transcript was identical to an annotated transcript (38 were identical to the transcript providing the 3′UTR, 3 to other isoforms of the gene). These genes were not considered modified. We use the polyester Bioconductor package (Frazee et al, 2015) (v1.16.0) to simulate approximately 1,000 strand-specific read pairs (read length 125 nt) from each of the 4,473 remaining artificial transcripts, and a total of 10 million read pairs distributed between 10,000 randomly selected transcripts, not annotated to any of the genes from which the artificial transcripts were generated. The simulated data set is then processed using the original Ensembl GRCh38.90 annotation files (which do not contain the artificial transcripts).

## Transcript abundance estimation

We use eight methods to estimate abundances of the annotated transcripts in each of the two Illumina libraries:

· RSEM. We build an index from the combined cDNA and ncRNA reference FASTA files from Ensembl and estimate transcript abundances with RSEM (Li & Dewey 2011) (v1.3.0), using bowtie (Langmead et al, 2009) (v1.1.2) as the underlying aligner.
· Salmon. We build a transcriptome index from the combined cDNA and ncRNA reference FASTA files from Ensembl and run Salmon (Patro et al, 2017) (v0.11.0) in quasi-mapping mode, incorporating sequence, GC, and positional bias correction. We also generate 100 bootstrap samples for estimation of the inferential variance for each transcript. By default, Salmon removes duplicated sequences in the reference catalog, keeping only one representative. In this process, 12,824 transcripts from 4,499 genes were excluded from the Ensembl GRCh38.90 catalog. In most of these cases, at least one of the identical sequences can be found on an alternative contig (e.g., in the MHC region). It's worth noting that these contigs are not included in the primary genome assembly used for the genomic alignments, whereas the transcripts are contained in the Ensembl transcriptome FASTA files. 3,450 of the affected genes did not have any other annotated transcript and were thus completely removed from the annotation catalog.
· SalmonKeepDup. Here, we run Salmon with the same settings as earlier, but retain all duplicated transcript sequences in the catalog (which is an option during Salmon's indexing step). Because the retained transcripts are sequence identical, the estimated abundances will be uniformly distributed within groups of identical transcripts.
· kallisto. We build a transcriptome index from the combined cDNA and ncRNA reference FASTA files from Ensembl and run kallisto (Bray et al, 2016) (v0.44.0) with bias correction activated.
· Hera. The Hera index is built using the reference genome (primary assembly) and the Ensembl gtf file, and Hera (https://github.com/bioturing/hera) (v1.1) is run with default settings.
· HISAT2+StringTie. We build a HISAT2 (Kim et al, 2015) (v2.1.0) index from the reference genome (primary assembly) and extract the known splice sites using the provided hisat2_extract_splice_sites.py script. The reads are aligned to this index with the option –dta set

and given the known splice sites. Next, we run StringTie (Pertea et al, 2015) (v1.3.3b) without assembly of new transcripts (-e option) to get the abundance estimates for the annotated transcripts.

· SalmonSTAR. For this approach, we build a transcriptome index from the combined cDNA and ncRNA reference files from Ensembl and align the reads using STAR (Dobin et al, 2013) (v2.5.3a). We subsequently estimate transcript abundances using Salmon (v0.11.0) in alignment-based mode, incorporating sequence and GC bias correction.

· SalmonCDS. Here, we build the Salmon index using only the explicitly annotated coding sequences from Ensembl, and run Salmon (v0.11.0) in quasi-mapping mode, incorporating sequence, GC, and positional bias correction.

### Prediction of expected junction coverage

To predict the expected number of reads mapping across each junction, given estimates of the transcript abundances, we first fit a fragment-level bias model using the alpine Bioconductor package (Love et al, 2016) (v1.2.0). The bias model is fit for each library separately, using a set of single-isoform genes with length between 600 and 7,000 bp and between 500 and 10,000 assigned reads. The alpine bias model includes random hexamer bias, fragment GC bias, positional bias along the transcript, and the fragment length distribution. After fitting the bias model, we use it to obtain a predicted coverage of each nucleotide in each annotated transcript using the fitted parameters for these four terms. For transcripts where the prediction fails (e.g., transcripts shorter than the estimated fragment length and transcripts with no overlapping reads), we assume a uniform coverage rather than excluding them from subsequent analysis steps. Next, we rescale the predicted base-level coverages by dividing with their total sum and multiplying with the average fragment length and the estimated transcript counts from each of the transcript abundance estimation methods to get an estimate of the number of reads predicted to cover each position on the transcript. We also extract the position of annotated splice junctions within each transcript, and the predicted coverage at the base just before an annotated junction is used as the predicted number of reads from that transcript that align across the junction. Finally, we sum the predicted number of junction-spanning reads for each junction across all transcripts, in a strand-aware fashion (because the libraries are stranded) to get the total number of reads predicted to span any given junction.

### Observed junction coverage

The observed junction coverage (the number of reads mapping across a given junction) is obtained using STAR (Dobin et al, 2013) (v2.5.3a). We build an index using the reference genome (primary assembly) and the Ensembl gtf file and align the reads with default settings. The number of uniquely mapping and multi-mapping reads spanning each annotated junction are extracted from the SJ.out.tab output file from the STAR alignment. Observed junction coverages can also be obtained by processing the bam file resulting from the genome alignment, for example, using Bioconductor packages such as QuasR (Gaidatzis et al, 2015) or GenomicAlignments. For our purposes, the advantage of the STAR output is that the numbers of uniquely mapping and multi-mapping reads spanning each junction are reported separately.

### The JCC score

To quantify the level of agreement between the predicted junction coverages based on any of the transcript abundance estimation methods and the observed number of junction reads from STAR, we define a family of gene-wise JCC scores, parametrized by two arguments: a weighting function $g$ and a scaling indicator $\beta$ (see the following equation). For a given $g$ and $\beta$, the JCC score for gene $i$ is defined by

$$JCC_i = \frac{\sum_{j \in J_i} g(\omega_j) \left| \left( \frac{\sum_{k \in J_i} g(\omega_k) R_k}{\sum_{k \in J_i} g(\omega_k) C_k} \right)^{\beta} C_j - R_j \right|}{\sum_{j \in J_i} g(\omega_j) R_j},$$

where $J_i$ denotes the set of junctions annotated to gene $i$ (some junctions are annotated to transcripts from multiple genes, in which case they are included for all of them), $R_j$ is the observed number of uniquely mapping reads spanning junction $j$ (obtained from STAR), and $C_j$ is the predicted number of reads spanning junction $j$ based on the bias model from alpine and the transcript abundances from a given method. Multi-mapping reads (from STAR) cause problems in the score calculation because it is not clear how to assign them to junctions, and thus the contribution of a junction is weighted by $g(\omega_j)$, where $g : [0, 1] \mapsto [0, \infty)$ is a non-negative function and $\omega_j$ is the fraction of reads spanning junction $j$ that are uniquely mapping.

Overall differences in the number of reads assigned to gene $i$ by transcript abundance estimation compared with junction counts can induce large differences between $C_j$ and $R_j$ even if their relative coverage patterns are similar. The same is true if there is a large fraction of multi-mapping reads, which are being included in the predicted transcript abundances but not in the observed junction coverages. To account for this, we include an optional scaling of the predicted coverages to have the same (weighted) sum as the observed coverages. This is represented by the $\beta$ parameter—if this is 0, no scaling is performed, and if it is 1, the values are scaled. In this study, we set $\beta = 1$, and let

$$g(\omega) = \begin{cases} 1 & \text{if } \omega \geq 0.75 \\ 0 & \text{otherwise} \end{cases},$$

that is, a step function that implies that only junctions with more than 75% uniquely aligning reads are allowed to contribute to the JCC score calculations. Overall, the JCC scores are robust to small changes in the weight function (Fig S30); in particular, the function only affects genes with a large number of multi-mapping reads.

With $\beta = 1$, which is the generally recommended setting, the JCC score for a gene takes values between 0 and 2. Without this scaling, multi-mapping reads can lead to large discrepancies between the observed and predicted junction coverages because these reads are typically contributing to the abundance estimates but not to the observed junction coverages. A low JCC score means that the predicted junction coverages, given the abundance estimates for the

transcripts in gene $i$, are compatible with the observed number of reads mapping across the junctions, whereas a high score indicates that for at least one junction, the predicted number of junction-spanning reads does not match with the observed number.

## Data access

Raw FASTQ files for the two Illumina libraries have been uploaded to ArrayExpress (accession number: E-MTAB-7089). All code used to perform the analyses is available from https://github.com/csoneson/annotation_problem_txabundance. An R package enabling calculation of the JCC score is available from https://github.com/csoneson/jcc.

# Supplementary Information

# Acknowledgements

The authors would like to thank the members of the Robinson, von Mering, and Baudis groups at the University of Zurich for helpful discussions. The authors would like to acknowledge the support from a Pilot Project grant from the University Research Priority Program Evolution in Action of the University of Zurich (to C Soneson), the National Science Foundation (BIO-1564917 and CCF-1750472 to R Patro), the National Human Genome Research Institute (R01HG009125 to MI Love), the National Cancer Institute (P01CA142538 to MI Love), the National Institute of Environmental Health Sciences (P30 ES010126 to MI Love), and Biotechnology and Biosciences Research Council UK (BB/N000749/1 to S Hussain).

## Author Contributions

C Soneson: conceptualization, data curation, software, formal analysis, funding acquisition, investigation, visualization, methodology, and writing—original draft, review, and editing.
MI Love: data curation, funding acquisition, methodology, and writing—review and editing.
R Patro: data curation, methodology, and writing—review and editing.
S Hussain: resources, data curation, funding acquisition, and writing—review and editing.
D Malhotra: resources, data curation, funding acquisition, and writing—review and editing.
MD Robinson: resources, data curation, supervision, funding acquisition, methodology, and writing—review and editing.

## Conflict of Interest Statement

The authors declare that they have no conflict of interest.

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
