## [Reviewer comments · Life Science Alliance]

Life Science Alliance

A junction coverage compatibility score to quantify the reliability of isoform abundance estimates

Charlotte Soneson, Michael Love, Rob Patro, Shobbir Hussain, dheeraj malhotra, and Mark Robinson

DOI: 10.26508/lsa.201800175

Corresponding author(s): Mark Robinson, University of Zurich

Review Timeline:

Submission Date:	2018-08-24
Editorial Decision:	2018-10-23
Revision Received:	2018-12-14
Editorial Decision:	2019-01-07
Revision Received:	2019-01-07
Accepted:	2019-01-08

Scientific Editor: Andrea Leibfried

Transaction Report:

October 23, 2018

Re: Life Science Alliance manuscript #LSA-2018-00175-T

Mark D Robinson
University of Zurich
Institute of Molecular Life Sciences
Winterthurerstrasse 190
IMLS
Zurich, ZH 8057
Switzerland

Dear Dr. Robinson,

Thank you for submitting your manuscript entitled "A junction coverage compatibility score to quantify the reliability of transcript abundance estimates and annotation catalogs" to Life Science Alliance. I apologize for the delay in getting back to you. I had to replace one of the reviewers at a very late stage because the reviewer became unresponsive, and this resulted in a longer reviewing time than usual. I have now received input from two experts, whose comments are appended to this letter.

As you will see, the reviewers appreciate your analyses, and they both support publication of a slightly revised version in Life Science Alliance. The reviewers make good suggestions on how to further strengthen the value of your analyses to others, and we would thus like to invite you to revise your work following the constructive input provided.

Thank you for this interesting contribution to Life Science Alliance. We are looking forward to

receiving your revised manuscript.

Sincerely,

- A letter addressing the reviewers' comments point by point.
- An editable version of the final text (.DOC or .DOCX) is needed for copyediting (no PDFs).
- High-resolution figure, supplementary figure and video files uploaded as individual files: See our detailed guidelines for preparing your production-ready images, <http://life-science-alliance.org/authorguide>
- Summary blurb (enter in submission system): A short text summarizing in a single sentence the study (max. 200 characters including spaces). This text is used in conjunction with the titles of papers, hence should be informative and complementary to the title and running title. It should describe the context and significance of the findings for a general readership; it should be written in the present tense and refer to the work in the third person. Author names should not be mentioned.

B. MANUSCRIPT ORGANIZATION AND FORMATTING:

Full guidelines are available on our Instructions for Authors page, <http://life-science-alliance.org/authorguide>

Reviewer #2 (Comments to the Authors (Required)):

1. In this manuscript, the authors present a methodology to assess the reliability of transcript expression estimation from RNASeq data without the need of gold standards or external validation. In order to attain this, the authors compute the expected number of reads covering each exon junction derived from the transcript expression and compare it to the observed number of reads for each annotated gene with at least one exon junction. Eight different methods are used to estimate expressions of two public and one simulated dataset. The authors conclude that although most methods have high concordance in most genes, some genes have poor scores and should be flagged for extra caution.

2. Overall the statements claimed in the paper are well supported by computation of statistics (e.g. correlation) or by descriptive plots showing the corresponding behaviour.

The authors notice that most genes show high compatibility between the observed and predicted junction coverages. This is true for most methods tested and the correlation improves when the counts are scaled within each gene. Furthermore, the correlation between the JCC score is high between methods, showing that most genes have similar scores regardless of the method for transcript quantification. The authors show examples of genes with poor JCC scores and give some insight about possible reasons for this behaviour.

I would suggest that the authors investigate further in three directions:

- How do different quantification methods compare for genes with high JCC scores? From Supp. Figures 8 and 9 it is clear that many methods give similar scores for all the range of possible values, nevertheless there are some clear discrepancies between some of them (e.g. SalmonCSD vs others or kallisto vs stringTie). Investigating the nature of these scores for dissimilar methods may give more information about the reason for bad performance of one or other method and the characteristics of genes with high scores. Also, general descriptives of these genes would be very informative, in terms of, for instance, how many of them suffer from the 3' end problems or whether there are other common reasons for poor performance.

- How is the JCC score affected by changes in the parameters of the defining function? It would be very informative for future users of the score to know how changes in these parameters affect the results and how to interpret different choices.

- What happens to the score when downsampling the number of reads per sample? Most quantification methods will perform similarly well for high coverages (such as those in the samples used in the manuscript) but will start failing with smaller number of reads. It would be interesting to know if a particular method performs better than other under different total number of reads. Equivalently, the computation of the JCC score will suffer from lower coverages and therefore conclusions could change depending on this parameter.

3. Specific comments:

Page 7 line 106: there is no description of how the "expected error" was computed. It is not straightforward to guess what it refers to.

Reviewer #3 (Comments to the Authors (Required)):

Soneson et. al. present an interesting study on how gene annotations can affect transcript-level quantification. They develop a JCC score to flag "problematic" genes, whose transcript-level quantifications

can be considered as unreliable.

The paper is very well written and the methodology presented in a sufficiently detailed way so I mainly have a couple of suggestions on how to improve the presentation of the results.

1.) The introduction of the JCC score in Figure 1B could be improved by emphasizing that it essentially compares observed to expected exon junction spanning read counts.

So I would suggest something like a two column layout for the figure with "observed" / "expected" columns and the alignment step on top:

1.) align

2a) observed | 2b) expected

I. observed junction counts | I. estimated transcript ab
| II. alpine bias model
| III. predicted junction counts

finally:

3.) JCC = observed vs expected junction counts

Maybe also choose different colors for steps 1, 2 and 3.

I think this would help the reader a lot to get a high-level conceptual overview of the score.

2.) The start of the results section is a bit add hoc:

"was done separately for each of the two Illumina libraries,"

The data studied only consists of Illumina libraries, so maybe it would be better to simply say "each of the samples in the two data sets used"

3.) Maybe it is useful to have hexbin plots for Figure 2A, Supplementary Figure 2, etc. to alleviate overplotting?

Exponential binning might be useful, see here (Figure 3.29) for an example:

<https://www.huber.embl.de/msmb/Chap-Graphics.html#rgraphics:sec:2d>

4.) The authors only report the number of genes affected by a high JCC score:

"146 genes in the Cortex library and 56 genes in the HAP1 library.
17 of the 168 genes pass the filters in both libraries."

without characterizing them further. I think this is a necessary addition to the paper as most of the readers will be very interested in the biological function of those genes.

So something similar to the "Characteristics of problematic genes" in the Robert and Watson paper would be a great addition.

5.) In supp Figure 4 and similar the description reads:
"The JCC scores for this gene based on the respectively abundances"

I am not sure the "respectively" is at the correct position here, but then I am not a native speaker.

This review was written by Bernd Klaus

A junction coverage compatibility score to quantify the reliability of transcript abundance estimates and annotation catalogs

Responses to reviewer comments

2018-12-14

We would like to thank the reviewers for their positive and constructive comments, which we feel have improved the manuscript. Below we provide point-by-point responses to the concerns raised by the reviewers (in *italics*).

Reviewer #2 (Comments to the Authors (Required)):

1. In this manuscript, the authors present a methodology to assess the reliability of transcript expression estimation from RNASeq data without the need of gold standards or external validation. In order to attain this, the authors compute the expected number of reads covering each exon junction derived from the transcript expression and compare it to the observed number of reads for each annotated gene with at least one exon junction. Eight different methods are used to estimate expressions of two public and one simulated dataset. The authors conclude that although most methods have high concordance in most genes, some genes have poor scores and should be flagged for extra caution.

2. Overall the statements claimed in the paper are well supported by computation of statistics (e.g. correlation) or by descriptive plots showing the corresponding behaviour.

The authors notice that most genes show high compatibility between the observed and predicted junction coverages. This is true for most methods tested and the correlation improves when the counts are scaled within each gene. Furthermore, the correlation between the JCC score is high between methods, showing that most genes have similar scores regardless of the method for transcript quantification. The authors show examples of genes with poor JCC scores and give some insight about possible reasons for this behaviour.

I would suggest that the authors investigate further in three directions:

- **How do different quantification methods compare for genes with high JCC scores?** From Supp. Figures 8 and 9 it is clear that many methods give similar scores for all the range of possible values, nevertheless there are some clear discrepancies between some of them (e.g. SalmonCSD vs others or kallisto vs stringTie). Investigating the nature of these scores for dissimilar methods may give more information about the reason for bad performance of one or other method and the characteristics of genes with high scores. Also, general descriptives of these genes would be very informative, in terms of, for instance, how many of them suffer from the 3' end problems or whether there are other common reasons for poor performance.

SalmonCDS and StringTie indeed show the most different JCC scores compared to the other methods. One explanation for this is illustrated in Figure 5, where we show that these two methods appear to put more weight on matching the internal structure (e.g. junctions) than the overall transcript sequence (including, e.g., UTRs) when assigning reads to transcripts. For SalmonCDS, this is not surprising, since the quantification is based on a different set of reference transcripts (only annotated coding sequences) than the other methods (see also Supplementary Figure 3, which shows the difference in the number of genes quantified by the different methods). Also for StringTie, it seems that the main reason for the discrepancy with the rest of the methods is that the quantification focuses on different aspects of the transcripts.

In the revised manuscript, we provide a more extensive investigation of the genes obtaining high JCC scores across all abundance estimation methods (Supplementary Figures 7-14). The rationale behind this selection is that genes that get high scores regardless of how the transcript abundances are estimated are more likely to be incompletely or incorrectly annotated, or to show a suboptimal read coverage, making abundance estimation difficult. In contrast, if a gene obtains a low JCC score for at least one set of abundance estimates, the provided transcript catalog can be compatible with the observed junction coverages, and any

high JCC scores are more likely to be due to imperfect abundance estimates rather than misannotation. In addition, we address the question of whether or not the observed high JCC scores are likely due to misannotated 3'UTRs, by investigating an additional annotation catalog containing a set of transcripts with extended 3'UTRs. Genes for which the JCC score improve consistently (for several abundance estimation approaches) with the extended transcript catalog indeed appear to have misannotated 3'UTRs (Supplementary Figures 20-29).

Explicitly investigating the biases of the different abundance estimation approaches, including what kind of imperfections in the annotation catalog that each method is most sensitive to, is indeed an interesting question, but a proper investigation of this is a large undertaking, and we feel that it goes beyond the scope of the current study. In fact, the main motivation for including multiple methods in the study was rather to illustrate that regardless of how the transcript abundances are estimated, there are genes with high JCC scores, and thus the issue can not be addressed by simply estimating the abundances with a different, "optimal", approach. Thus, our aim is also not to compare the methods in terms of "performance" - this has been done before in setups where the true abundances are known.

- How is the JCC score affected by changes in the parameters of the defining function? It would be very informative for future users of the score to know how changes in these parameters affect the results and how to interpret different choices.

The choice of parameters is, ultimately, up to the researcher, deciding whether to allow multimapping reads to influence the calculated scores or not. In the revised manuscript, we compare JCC scores calculated with five different weight functions g (Supplementary Figure 30). Overall, the scores are not strongly affected by the choice of g (in addition, it only affects the genes with a large number of multimapping reads). Comparing estimates obtained with different choices of β is considerably more difficult, since with $\beta=0$ the scores are no longer confined to the interval $[0,2]$. In addition, without the scaling obtained by β , multimapping reads will cause a large difference between the observed and predicted number of reads spanning a junction (since these reads are typically included in the transcript abundance estimates). We have clarified this in the revised manuscript.

- What happens to the score when downsampling the number of reads per sample? Most quantification methods will perform similarly well for high coverages (such as those in the samples used in the manuscript) but will start failing with smaller number of reads. It would be interesting to know if a particular method performs better than other under different total number of reads. Equivalently, the computation of the JCC score will suffer from lower coverages and therefore conclusions could change depending on this parameter.

Since the (mis)annotations in the transcript catalog are independent of the reads themselves, the main effect of changing the number of reads will be on the ability of the methods to estimate transcript abundances, and on the number of genes for which the JCC score can be calculated. While, as also mentioned above, we consider an investigation of the "performance" of the individual methods (such as the ability to estimate abundances with reduced read numbers) to be beyond the scope of this paper, this question can be partly addressed by comparing the two real data sets (HAP1 with ~55M reads, Cortex with ~110M). As can be seen in Supplementary Figure 3, more genes are covered to a sufficient depth to calculate the JCC score in the Cortex sample (dark blue bars). However, the agreement between the scores from different methods are generally comparable in the two data sets (Supplementary Figures 4-5).

3. Specific comments:

Page 7 line 106: there is no description of how the "expected error" was computed. It is not straightforward to guess what it refers to.

Thanks for pointing this out. This sentence has been clarified, and now reads "the prediction of the coverage pattern by alpine failed for 29,342 (14.6%) in the HAP1 sample and 13,906 (6.9%) in the Cortex sample, almost exclusively due to transcripts being shorter than the respective fragment lengths". In fact, we did not compute an "expected error"; instead, what we intended to say was that alpine failed to predict transcript coverages for 7-15% of the transcripts, since the fragment lengths were longer than the transcripts. This failure was "expected", since there is no way for alpine to determine the coverage profile in these cases.

Reviewer #3 (Comments to the Authors (Required)):

Soneson et. al. present an interesting study on how gene annotations can affect transcript-level quantification. They develop a JCC score to flag "problematic" genes, whose transcript-level quantifications can be considered as unreliable. The paper is very well written and the methodology presented in a sufficiently detailed way so I mainly have a couple of suggestions on how to improve the presentation of the results.

1.) The introduction of the JCC score in Figure 1B could be improved by emphasizing that it essentially compares observed to expected exon junction spanning read counts.

So I would suggest something like a two column layout for the figure with "observed" / "expected" columns and the alignment step on top:

1.) align

2a) observed | 2b) expected

I. observed junction counts | I. estimated transcript ab

| II. alpine bias model

| III. predicted junction counts

finally:

3.) JCC = observed vs expected junction counts

Maybe also choose different colors for steps 1, 2 and 3. I think this would help the reader a lot to get a high-level conceptual overview of the score.

Thanks for the suggestion. In the revised manuscript, we have restructured Figure 1B according to your comments.

2.) The start of the results section is a bit ad hoc:

"was done separately for each of the two Illumina libraries,"

The data studied only consists of Illumina libraries, so maybe it would be better to simply say "each of the samples in the two data sets used"

This sentence has been rephrased to "It is done separately for the HAP1 and Cortex samples, in order to account for any sample-specific biases."

3.) Maybe it is useful to have hexbin plots for Figure 2A, Supplementary Figure 2, etc. to alleviate overplotting? Exponential binning might be useful, see here (Figure 3.29) for an example:

<https://www.huber.embl.de/msmb/Chap-Graphics.html#rgraphics:sec:2d>

Thank you for the suggestion. While hexbin plots would indeed be helpful to illustrate the density in different parts of the plot, we finally decided that we would like to keep the current plots, in order to be able to highlight the junctions with a large number of multimapping reads in a different color.

4.) The authors only report the number of genes affected by a high JCC score:

"146 genes in the Cortex library and 56 genes in the HAP1 library. 17 of the 168 genes pass the filters in both libraries."

without characterizing them further. I think this is a necessary addition to the paper as **most of the readers will be very interested in the biological function of those genes.**

So something similar to the "Characteristics of problematic genes" in the Robert and Watson paper would be a great addition.

In the revised manuscript, we have further investigated the genes obtaining a high JCC score with all the methods, and categorized them into three groups: those where the cause of the high score seems to be a poor coverage in the 3'/5' end, those where there appears to be a missing transcript (or a too short 3'UTR), and those with more complex behaviour (Supplementary Figures 7-14). We also note that in practice, these genes will not necessarily get a high score in all data sets; that will ultimately depend on which transcripts are actually expressed. Thus, while a list of "difficult" genes can be informative, they will not necessarily be problematic in all data sets (nor will we have detected all of them in these two data sets), and thus should not always be treated with special care. This is somewhat different from the problematic genes found by Robert and Watson, where the shared sequence will make abundance estimation ambiguous in any data set. (Note that due to an update of one of the evaluated methods, the number of shared high-scoring genes have changed slightly in the revised manuscript).

5.) In supp Figure 4 and similar the description reads: "The JCC scores for this gene based on the respectively abundances"
I am not sure the "respectively" is at the correct position here, but then I am not a native speaker.

Thanks for noticing this, it has been corrected.

This review was written by Bernd Klaus

January 7, 2019

RE: Life Science Alliance Manuscript #LSA-2018-00175-TR

Prof. Mark D Robinson
University of Zurich
Institute of Molecular Life Sciences
Winterthurerstrasse 190
IMLS
Zurich, ZH 8057
Switzerland

Dear Dr. Robinson,

Thank you for submitting your revised manuscript entitled "A junction coverage compatibility score to quantify the reliability of isoform abundance estimates". As you will see, the reviewers appreciate the introduced changes and we would thus be happy to publish your paper in Life Science Alliance pending final revisions necessary to meet our formatting guidelines:

- please include the supplementary figure legends in the main manuscript text file and upload the S figures without legends and as individual files (one pdf page per figure).

A. FINAL FILES:

-- High-resolution figure, supplementary figure and video files uploaded as individual files: See our detailed guidelines for preparing your production-ready images, <http://life-science-alliance.org/authorguide>

B. MANUSCRIPT ORGANIZATION AND FORMATTING:

Full guidelines are available on our Instructions for Authors page, <http://life-science-alliance.org/authorguide>

Sincerely,

Reviewer #2 (Comments to the Authors (Required)):

I apologise for the late reply. The authors have addressed all my concerns. I have no further requests.

Reviewer #3 (Comments to the Authors (Required)):

The authors have nicely incorporated my suggestions and addressed the concerns raised, so I don't have any further comments.

January 8, 2019

RE: Life Science Alliance Manuscript #LSA-2018-00175-TRR

Prof. Mark D Robinson
University of Zurich
Institute of Molecular Life Sciences
Winterthurerstrasse 190
IMLS
Zurich, ZH 8057
Switzerland

Dear Dr. Robinson,

Thank you for submitting your Research Article entitled "A junction coverage compatibility score to quantify the reliability of isoform abundance estimates". It is a pleasure to let you know that your manuscript is now accepted for publication in Life Science Alliance. Congratulations on this interesting work.

DISTRIBUTION OF MATERIALS:

Again, congratulations on a very nice paper. I hope you found the review process to be constructive and are pleased with how the manuscript was handled editorially. We look forward to future exciting

submissions from your lab.

Sincerely,
